# The gender gap in STEM: (Female) teenagers' ICT skills and subsequent career paths

**Friederike Hertweck**[1]*, **Judith Lehner**[2]

**1** RWI - Leibniz Institute for Economic Research, Essen, Germany, **2** University of Bayreuth, Bayreuth, Germany

\* friederike.hertweck@rwi-essen.de

**Data Availability Statement:** Data is available for free from the Leibniz Institute for Educational Trajectories upon completing a data user agreement. URL: https://www.neps-data.de/Data-Center/Data-and-Documentation/Start-Cohort-

## Abstract

Skills shortage in the fields of Sciences, Technology, Engineering, and Mathematics (STEM) poses a significant challenge for industries globally. To overcome shortage of STEM talent, the selection into STEM fields must be fully understood. We contribute to existing research on the selection of STEM careers by analysing the interplay between teenagers' proficiency in Information and Communication Technology (ICT) and their career preferences in the STEM domain. Based on representative data for German teenagers, our study shows that female teenagers are less likely to choose a career in STEM unless they have strong ICT skills in secondary school. The relationship does not hold for male students. An increase in girls' ICT skills by 10 percentage points in ninth grade is associated with an increase in the probability to choose a STEM career by 2.95 percentage points. Our findings can be explained with evidence that teenagers sort into occupations they believe to be good at and that female teenagers rather underestimate their true potential. Using different empirical approaches, we also show that ICT skills act as a moderator and not as a mediator in the gender-specific choice of training upon graduating from secondary school. By addressing the interplay between gender, ICT skills, and educational choices, the present study uncovers an additional lever of how to mitigate skills shortage in STEM.

## Introduction

Many companies and industries are struggling to find qualified candidates to fill open positions in sciences, technology, engineering, and mathematics (STEM). This shortage of STEM talent is due to the persistent gender gap in STEM, the lack of investment in STEM education, inadequate training, and the high level of competition for top talent [1–4]. To overcome skills shortage in STEM, more teenagers in general and female youth in particular must choose a STEM occupation after completing high school.

This paper analyses the interplay between teenagers' gender, digital skills, and enrollment in STEM fields upon graduating from high school. The study is based on the German National Educational Panel Study (NEPS) that combines data from testing teenagers' skills in Information and Communications Technology (ICT skills) with their subsequent educational trajectories.

Grade-9/Data-and-Citation DOI: https://doi.org/10.5157/NEPS:SC4:13.0.0.

**Funding:** Financial support was provided by the Ministry of Culture and Science of the state of North-Rhine Westfalia and by the Federal Ministry for Economic Affairs. Furthermore, the publication of this article was funded by the Open Access Fund of the Leibniz Association. The funders had no role in study design, data collection and analysis, decision to publish, or preparation of the manuscript.

**Competing interests:** The authors have declared that no competing interests exist.

We use a panel of 9,315 teenagers who were first tested in 2010 during ninth grade and were repeatedly asked about their educational and occupational choices. In a set of linear probability models, we regress educational and occupational choices during the five years after graduating from high school on students' ICT skills and a large set of socio-economic characteristics. Moreover, we perform moderation and mediation analyses to understand the interplay between gender, ICT skills, and choice of STEM fields.

Our results show that girls are much more likely to choose a STEM field upon graduating from high school only if they have strong ICT skills in ninth grade. This relationship is not present for boys. More precisely, an increase in girls' ICT skills by 10 percentage points in ninth grade is associated with an increase in the probability to choose a STEM field upon graduating from high school by 2.95 percentage points or around 25%. ICT skills in grade 9 moderate the gender-specific sorting into STEM fields. We explain this finding with existing evidence from psychological research that shows that teenagers rather choose occupations that span tasks they believe to be good at [5] and that female teenagers rather underestimate their true potential [6–8].

Moreover, we find that higher ICT skills close to graduating from high school (i.e., in grade 12) are positively associated with choosing a STEM field—irrespective of gender. A potential underlying reason for the positive association is that students in upper secondary school have to specialize in some fields and, as [1, 9] show, the selection into or out of STEM already partially takes place in high school. Thereby, our results complement recent evidence provided by [1, 9]. We show that the selection out of STEM fields is at least partly driven by teenagers' ICT skills in ninth grade and that girls sort into STEM fields upon graduating from high school only if they have above average ICT skills.

Existing studies that link gender-specific sorting into STEM careers and specific skills usually concentrate on (self-perceived) skills in mathematics or science. Reading skills are usually not adressed, most likely because the gender gap in reading skills favors girls [10]. Regarding mathematical skills, [11] show that mathematical skills and self-perceived mathematical ability positively affect the probability of majoring in science in college. Hence, the gender gap in mathematical skills, that arises as early as elementary school in favor of boys [10] and is strong among the upper part of the achievement distribution [12], may lead to the gender-specific sorting into careers.

Similarly, [13, 14] demonstrate that male high school students tend to possess a greater sense of self-perceived mathematical proficiency compared to their female counterparts. Furthermore, these elevated self-perceived abilities exhibit a positive correlation with the likelihood of selecting mathematics courses during high school and pursuing majors in math-intensive fields in college. Remarkably, this correlation persists even when adjusting for their actual mathematical abilities. However, forcing girls into mathematics at high school widens the gender gap in self-perceived mathematical skills and enrollment in STEM despite shortening the gap in mathematical skills [15, 16].

Additional lines of research focused on understanding the persistent gender gap in STEM examine attitudes towards STEM, peer effects, and role models. For instance, female students' perception that a college major is rather male dominated [17] as well as female students' more pronounced stereotypes when it comes to computer scientists or engineers [18] may hinder these students from choosing STEM [19]. In turn, being exposed to *high-performing female peers* in mathematics increases the probability to choose a science track in high school [20], while being exposed to a larger proportion of female peers or high-ability peers rather reduces the choice of a STEM major or occupation [21, 22].

Overall, existing evidence summarises a variety of factors why females are still underrepresented in STEM. Regarding a student's skills, the majority of existing studies concentrates on

mathematical skills. An overlooked but given the technological change increasingly important lever are ICT skills. To explore the potential links between gender, ICT skills, and occupational choice, we employ different empirical methods to understand whether ICT skills moderate or mitigate the gender-specific sorting into STEM: It is possible that ICT skills affect field choices entirely independent of gender, especially if stereotypes were absent. However, [18] show that female students' have more pronounced stereotypes regarding computer scientists than their male peers. As a result, girls may be less willing to build up ICT skills so that subsequent career choices are directly affected by a student's gender and a student's ICT skills but also indirectly by the connection between gender and ICT skills. A priori it is therefore not clear how gender, ICT skills, and career choices are interrelated.

By testing whether and how teenagers' ICT skills affect their subsequent fields of training, this study helps identify potential intervention points for addressing gender disparities in STEM fields. We show that ICT skills in ninth grade affect long-run educational decisions. Promoting ICT skills and confidence from secondary school onwards especially among female youth can boost enrollment in STEM and may combat the persistent gender gap in STEM. This finding is of particular importance because recent research on the gender gap in ICT skills shows that gender-specific difference in digital skills is negligibly small among teenagers and emerges at age 18 in favor of boys [23–25]. Notwithstanding these small differences, girls frequently opt out of STEM-related fields in upper secondary school [1], most likely due to a lack of academic confidence in STEM subjects [26].

By addressing the interplay between gender, ICT skills, and educational choices, the present study uncovers an additional lever of how to mitigate skills shortage in STEM. The findings of this study thereby also add to the ongoing policy debate on increasing the intensity of ICT training in secondary school [27, 28]. By fostering ICT skills and strengthening confidence in ICT, especially among teenager girls in secondary education, the likelihood of more teenagers pursuing STEM careers can be enhanced. This approach may effectively address the enduring gender disparity in STEM fields.

## Data and methods

ICT skills are often labelled as *ICT literacy* or *digital literacy* and have been initially described as "the ability to understand and use information in multiple formats from a wide range of sources when it is presented via computers" [29, p. 1]. However, since the required digital competencies are subject to continuous technological change, it is difficult to find a universally valid definition of the term over a long period of time [30, 31].

In 2002, the Educational Testing Service (ETS) established the ICT Literacy Panel and defined ICT Literacy as "using digital technology, communications tools, and/or networks to access, manage, integrate, evaluate, and create information in order to function in a knowledge society" [32, p. 2]. These five components—access, manage, integrate, evaluate, and create—have subsequently been used in this (or slightly modified) form in numerous studies to describe ICT skills.

For instance, [31] find most of these five components in all twelve peer-reviewed definitions from national and international organizations, including OECD and UNESCO. In their studies on ICT skills, [33–36] also base their measurements on these components. Some studies also address ethical and legal issues relating to the use of information on the Internet [31, 34, 37] or add the time needed to solve an ICT problem as an additional dimension [38].

### Description of the data

The present study is based on the German National Educational Panel Study (NEPS; see [39]). The NEPS is a representative multi-sequence study designed to research educational

trajectories over the life course. It is carried out by the Leibniz Institute for Educational Trajectories (LIfBi) in Germany, in cooperation with a nationwide network. All data collection procedures, instruments and documents were checked by the data protection unit of the Leibniz Institute for Educational Trajectories (LIfBi).

In Germany, teenagers can attend different school tracks that provide the possibility to start vocational training after completing the 9th or 10th grade, or to pursue a university entrance qualification, typically after the 12th or 13th grade. The NEPS allows to link data on teenagers' educational trajectories to their performance in skills tests.

ICT skills are tested among kids, teenagers, college students, and adults. The NEPS takes up four of the aforementioned process components—access, manage, evaluate, and create—in the design of the ICT skill tests. Each component covers both technological and cognitive aspects [40, 41].

Because we are interested in establishing the (gender-specific) link between teenagers' ICT skills and their subsequent career paths, we concentrate our analysis on a panel of teenagers who were first surveyed in the year 2010 when they attended ninth grade of high school [39]. Since then, the respondents have been repeatedly asked about their lives. Hence, the panel dataset comprises data on each student's educational and occupational choices including the type and field of training. We can only distinguish between female and male teenagers in the data.

In 2010, the teenagers participated in ICT skill tests. The majority of these teenagers had an additional test in the school year 2013/14 during 12th grade shortly before graduating from high school. Fig 1 illustrates the timeline.

During the ICT skill test in ninth grade (12th grade), respondents had to answer 39 (32) multiple choice items of different complexity that measured the extent to which they were able to access, manage, evaluate, and create digital content using various software application groups including word processing, spreadsheet, presentations, e-mail, Internet, and search engines [41]. The tests were paper-based and contained pictures of screenshots of standard software applications.

The NEPS test design comprises both simple and complex multiple-choice questions and the majority of respondents answered all questions [42]. It may be argued, however, that the tests may rather capture the stock of knowledge and not necessarily the practical skills. However, a comparison between NEPS and the simulation-based items of the International Computer and Information Literacy Study (ICILS) shows that the NEPS in fact provides a good mapping of digital skills [31, 43].

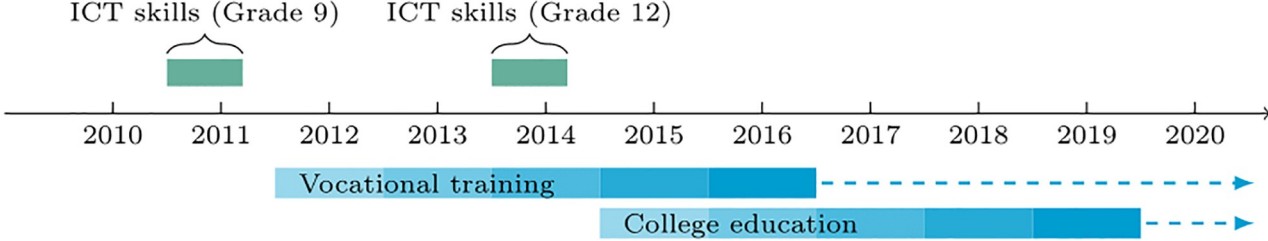

**Fig 1. Timeline of ICT skills tests and training choices.** The figure illustrates the timeline of events extracted from NEPS data. Secondary school students are tested in grade 9 (school year 2010/11) and some also again in grade 12 (school year 2013/14). Tests are indicated by the green boxes. Teenagers who are interested in vocational training can start the training after grade 9 at the earliest, though most students start after grade 10 or even later. Teenagers who intend to go to college must receive their university entrance qualification before enrolling. Hence, starting college is possible in 2014 at the very earliest. For each individual we consider a five-year time window after leaving secondary school as indicated by the blue bars. Potential differences in the exact timing of the five-year window are indicated by the arrows.

## Development of a score of ICT skills

To summarize the large set of up to 39 items of the ICT skills tests, we construct a weighted score of each respondent's digital skills. In doing so, we scale all items $k_f$ between 0 and 1 and add the number of correctly answered items relative to the total number of items queried:

$$ICTskills_i = \frac{1}{F} \sum_{f=1}^{F} k_f \qquad (1)$$

Items that remained unanswered or received an incorrect response receive a value of 0. The resulting score, $ICTskills_i$, lies between 0 to 1. A respondent's score of, for instance, 0.5 indicates that half of the queried items had been answered correctly. One correctly answered item increases the ICT skills by 0.026 in ninth grade and by 0.032 in 12th grade.

The NEPS provides alternative measures for ICT skills by scaling the test results using models of Item Response Theory to account for the fact that ICT skills are an unobserved characteristic of a person. In the data, weighted maximum likelihood estimates (WLE) and plausible values are provided. A respondent's WLE score is a point estimate that expresses the most likely competence score for each single respondent given their item responses. Plausible values are multiple imputations for the latent variable in an item response model. Please refer to [44, 45] for more details on the scaling of the results from the ICT skills test in grades 9 and 12, respectively.

All three measures—our mean score as well as the provided WLE score and plausible values—are based on a respondent's test scores. For ease of interpretation of regression results, we utilize the mean score from Eq 1 in our regressions. We also run regressions using the WLE scores. The direction of the effects remains the same, the statistical significance is similar—yet, the coefficients are easier to interpret when using the mean score as in Eq 1.

## Definition of STEM fields

The data contain information on vocational training and college education. Based on the five-digit occupation codes (KldB 2010) and the classification of STEM occupations by the German Federal Employment Agency, we can classify each vocational training as STEM or non-STEM. The Federal Employment Agency provides an exhaustive list of all occupations that require a high proportion of knowledge from the fields of Mathematics, Computer Science, Natural Sciences and/or Technology [46].

For those respondents who enrolled to college, the NEPS data contain information on the field of study. We classified fields as STEM in line with the subject classification of the German Federal Statistical Office [47]. More precisely, all fields of the subject groups "Mathematics and Natural Sciences" and "Engineering" were classified as STEM.

## Sample selection

We restrict our analysis to teenagers who were tested in ninth grade and participated in subsequent waves of the panel survey in which educational and occupational choices were surveyed. Moreover, we exclude all students from our analysis of which data on parents or the student's gender or migration background were missing. The relevant information is available for 57% of the data.

Our sample spans data on 9,315 students whose ICT skills were tested in ninth grade in 2010. Table 1 provides summary statistics of the sample. Among the 9,315 students in the sample, 50.1% are female and 21.9% have a migration background. Around one third of the

**Table 1. Descriptive statistics of the sample.**

| Statistic | Overall | | Male | | Female | |
|---|---|---|---|---|---|---|
| | **Mean** | **St. Dev.** | **Mean** | **St. Dev.** | **Mean** | **St. Dev.** |
| ICT skills in 9th grade | 0.569 | 0.174 | 0.565 | 0.179 | 0.573 | 0.169 |
| Female | 0.501 | 0.500 | – | – | – | – |
| Migration Background | 0.219 | 0.414 | 0.210 | 0.407 | 0.229 | 0.420 |
| Parent(s) in STEM Occupation | 0.337 | 0.473 | 0.347 | 0.476 | 0.326 | 0.469 |
| Mathematical Skills | 0.499 | 0.500 | 0.564 | 0.496 | 0.435 | 0.496 |
| Choose STEM: longest training | 0.326 | 0.469 | 0.513 | 0.500 | 0.140 | 0.347 |
| Choose STEM: first training | 0.337 | 0.473 | 0.526 | 0.499 | 0.149 | 0.356 |
| Choose STEM: last training | 0.317 | 0.465 | 0.501 | 0.500 | 0.134 | 0.341 |

The table provides the mean and standard deviation of variables of interest. The overall sample consists of 9,315 teenagers for whom all variables are available. The subsamples contain 4,747 female and 4,668 male teenagers. The overall sample is slightly positive selected—please see S1 Table for a comparison of the (representative) full data and the sample. Please also note that the variable describing mathematical skills is an indicator variable that equals 1 if students have above-average mathematical skills in grade 9 and 0 otherwise. We define above-average skills as having a WLE score above 0. Please refer to [48] for a detailed description of the competence tests in mathematics and the estimation of the WLE scores.

teenagers (33.7%) have at least one parent who works in a STEM occupation. NEPS SC4 is representative of German high school students. S1 Table shows that our sample is slightly positive selected in terms of ICT skills and overall mathematical skills. The sample comprises fewer students with a migration background but more students whose parents work in STEM —compared to the full NEPS data. We will discuss in the results section how the slightly positive selection may affect the results.

The average *ICT skills* is 0.569, meaning that the average respondent correctly answered 22 out of 39 questions (56.9%). Female teenagers have on average slightly higher ICT skills than their male fellows. Table 1 already reveals some stereotypical career choices: More than half of the male teenagers choose a career in STEM while less than 15% of the female teenagers go into STEM after completing secondary school.

Teenagers can start and drop out of their educational choices. We therefore include three different possibilities to measure whether a student chooses a STEM occupation after graduating from high school. Within five years after completing high school, we can distinguish between the *longest* training period, the *first* training period, i.e., the training period that starts directly upon graduating from secondary schooland, and the *last* training period, i.e., the one closest to the end of the five-year time window of interest. The longest training period is the training period with the longest duration, i.e., the training period that usually has been completed successfully. Table 1 shows that one third of the teenagers (33.7%) choose a STEM occupation directly after completing high school (*Choose STEM: first training*). It also shows that slightly fewer students have STEM as their longest (32.6%) or last training period (31.7%).

## Empirical analysis

We are interested in understanding the interplay between teenagers' digital skills and gender-specific selection into STEM fields upon graduation from secondary school. The study's objective is to address the following research questions:

1. Do teenagers' ICT skills correlate with their subsequent career choices and, if yes, to what extent? (RQ1)

2. Do differences exist in ICT skills and career choices of female and male teenagers? (RQ2)

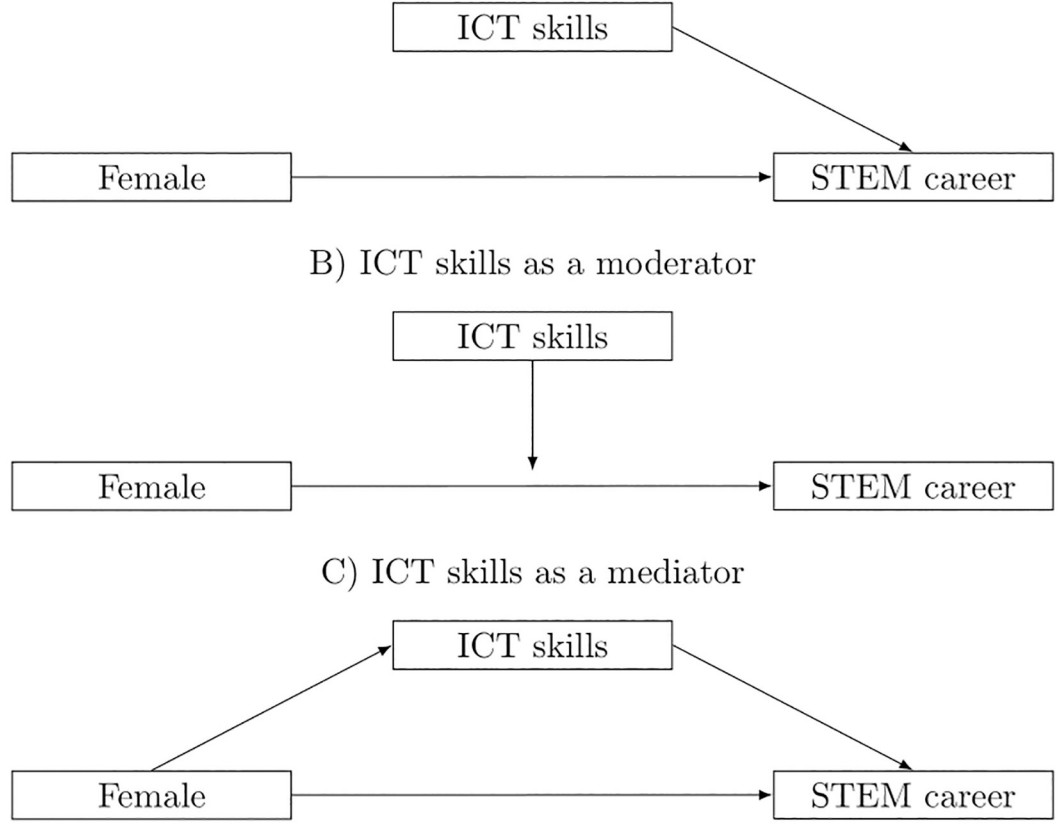

**Fig 2. Illustration of statistical models.** Panels A, B, and C illustrate stylized versions of the three possible scenarios of how gender, ICT skills, and the decision towards a career in STEM can relate to each other. The models in panels A and B are estimated by utilizing linear probability models. The model in panel C is estimated by running a mediation analysis. The full models that include all variables are illustrated in S1 Fig.

3. Is the gender-specific selection into STEM fields mitigated by ICT skills in secondary school? (RQ3)

As we do not know a priori whether a teenager's ICT skills affect subsequent career choices at all or act as a moderator or effect modifier, we employ three different statistical models. These are illustrated in Fig 2 and are described in more detail below.

**Direct relationship between ICT skills and STEM career.** To answer research question RQ1, we estimate a set of linear probability models with the method of ordinary least squares that postulate independent links between gender and ICT skills (Panel A in Fig 2). Our outcome variable $y_i$ is a binary variable that equals one if the respondent chooses a STEM major at college or a vocational training in a STEM field. More specifically, we estimate

$$P(y_i = 1 | X_{ij}) = \alpha + \beta ICTskills_i + \sum_{j=1}^{J} \gamma_j X_{ij} + \epsilon_i, \tag{2}$$

where $ICTskills_i$ captures the digital literacy of a respondent $i$ as described by Eq 1. $X_{ij}$

comprises a set of *j* individual control variables including a respondent's gender, migration background, and whether at least one of the parents worked in a STEM occupation. Moreover, $X_{ij}$ includes a binary variable for an individual's mathematical skills. It is 0 if students have below-average test results in mathematics in grade 9 and 1 otherwise. The data also include estimates for mathematical competences in the form of weighted maximum likelihood estimates (WLE scores). We use these WLE scores from the competence tests in mathematics that took place in grade 9. Please see [48] for a detailed description of the data on mathematical competence and the scaling model applied to estimate the WLE scores.

Our main specification utilizes the longest training period within five years after having completed secondary education. In a set of robustness checks, we change this definition from the longest to the first and to the last training period within five years after secondary schooling.

**ICT skills as a moderator.** To answer RQ2, we add interactions between the ICT skills and gender to Eq 2. The interaction term helps us understand whether the gender-specific sorting into STEM occupations is moderated by ICT skills (Panel B in Fig 2). Formally, we estimate

$$P(y_i = 1 | X_{ij}) = \alpha + \beta_1 ICTskills_i + \beta_2 ICTskills_i \cdot gender_i + \sum_{j=1}^{J} \gamma_j X_{ij} + \epsilon_i. \qquad (3)$$

By including the interaction term between ICT skills and gender as formalized in Eq 3, we can differentiate whether ICT skills directly affect the probability to choose a STEM career (provided by $\beta_1$) or whether gender-specific differences in these skills affect the relationship (provided by $\beta_2$).

**Mediation analysis: Understanding a potential indirect effect.** In a third set of regressions, we address RQ3 and aim at understanding whether the gender-specific selection into STEM fields is mitigated by ICT skills. We further explore the triangle between gender, ICT skills, and career choices by employing mediation analysis (Panel C in Fig 2). Mediation analysis can help unravel the underlying causal mechanism through which an explanatory variable affects an outcome into an effect running through changes of an intermediate variable and through other pathways. In our setting, as illustrated in Panel C in Fig 2, we are interested in the mediating effect of ICT skills on the gender-specific selection of STEM fields. The goal of the analysis in Panel C is to disentangle the total effect of gender on subsequent career paths into a direct effect and an indirect effect via a student's ICT skills. Thereby, the potential mediator—ICT skills—can help us clarify the mechanisms underlying the relationship between teenagers' ICT skills, gender, and career choices.

## Results

Table 2 provides the regression results from running linear probability models as formalized in Eqs 2 and 3. The dependent variable is always a binary variable that equals 1 if a teenager chooses college or vocational training in a STEM field after secondary schooling, and 0 otherwise. Columns 1a and 1b of Table 2 show that those teenagers rather choose a career in STEM that have high ICT skills (column 1a) or are male (column 1b). The coefficient for ICT skills in column 1a can be interpreted as follows. If a teenager's ICT skills increase by 10 percentage points (i.e., correctly answering an additional of three to four questions in the ICT skills test), the probability to choose a STEM field increases by 1.49 percentage points or 4.57%.

**Table 2. ICT skills in 9th and 12th grade and career paths: Direct relationship and moderating effect.**

| | Dependent variable: | | | | | | | | | |
|---|---|---|---|---|---|---|---|---|---|---|
| | Respondent chooses STEM occupation after high school | | | | | | | | | |
| | ICT skills in 9th grade | | | | | | ICT skills in 12th grade | | | |
| | (1a) | (1b) | (1c) | (1d) | (1e) | (1f) | (2a) | (2b) | (2c) | (2d) |
| ICT skills | 0.149*** | - | 0.158*** | 0.037 | −0.046 | −0.016 | 0.305*** | 0.308*** | 0.205* | 0.154 |
| | (0.028) | - | (0.026) | (0.035) | (0.038) | (0.041) | (0.067) | (0.094) | (0.094) | (0.096) |
| Female | - | −0.373*** | −0.372*** | −0.519*** | −0.509*** | −0.496*** | −0.264*** | −0.259** | −0.191* | −0.184* |
| | - | (0.009) | (0.009) | (0.030) | (0.030) | (0.031) | (0.017) | (0.079) | (0.079) | (0.079) |
| ICT skills * Female | - | - | - | 0.258*** | 0.256*** | 0.194*** | - | −0.008 | −0.077 | 0.024 |
| | - | - | - | (0.051) | (0.051) | (0.059) | - | (0.132) | (0.131) | (0.137) |
| Migration Background | - | - | −0.014 | −0.012 | −0.008 | −0.008 | −0.022 | −0.022 | −0.010 | −0.010 |
| | - | - | (0.011) | (0.011) | (0.011) | (0.011) | (0.023) | (0.023) | (0.023) | (0.023) |
| Parent(s) in STEM | - | - | 0.076*** | 0.076*** | 0.073*** | 0.074*** | 0.075*** | 0.075*** | 0.070*** | 0.069*** |
| | - | - | (0.009) | (0.009) | (0.009) | (0.009) | (0.018) | (0.018) | (0.018) | (0.018) |
| Mathematical Skills | - | - | - | - | 0.060*** | 0.038*** | - | - | 0.131*** | 0.193*** |
| | - | - | - | - | (0.010) | (0.015) | - | - | (0.019) | (0.031) |
| Mathematical Skills * Female | - | - | - | - | - | 0.043** | - | - | - | −0.098* |
| | - | - | - | - | - | (0.021) | - | - | - | (0.039) |
| Constant | 0.241*** | 0.513*** | 0.400*** | 0.468*** | 0.481*** | 0.476*** | 0.282*** | 0.280*** | 0.242*** | 0.225*** |
| | (0.017) | (0.006) | (0.017) | (0.021) | (0.022) | (0.022) | (0.043) | (0.058) | (0.058) | (0.058) |
| Observations | 9,315 | 9,315 | 9,315 | 9,315 | 9,315 | 9,315 | 2,789 | 2,789 | 2,789 | 2,789 |
| $R^2$ | 0.003 | 0.158 | 0.168 | 0.171 | 0.174 | 0.174 | 0.099 | 0.099 | 0.114 | 0.116 |
| Adjusted $R^2$ | 0.003 | 0.158 | 0.168 | 0.170 | 0.173 | 0.173 | 0.098 | 0.097 | 0.112 | 0.114 |

The dependent variable is a binary variable that equals 1 if the respondents' *longest* training period within five years after completing secondary schooling was within a STEM field. The ICT skills relate to the tests conducted in 9th grade (columns 1a to 1f) and 12th grade (columns 2a to 2d). Due to panel attrition, the subset of students who were again tested in 12th grade was much smaller. In columns 2a to 2d we only consider the selection into a STEM field after grade 12. Robust standard errors are clustered on the levels of schools.

Significance:

*$p < 0.1$;

**$p < 0.05$;

***$p < 0.01$.

In an additional set of regressions, we run the set of linear probability models outlined in Eqs 2 and 3 on the subsample of teenagers who stayed in high school and participated in another test on ICT skills in 12th grade. The subsample comprises 2,789 students. Compared to our baseline sample that has been surveyed in ninth grade we now have more female students and fewer students with a migrant background. The complete descriptive statistics are in S2 Table.

## Direct relationships

In columns 1c and 2a, we provide the results of jointly including ICT skills and gender in the regression as provided by Eq 2. A higher digital literacy as well as being male is associated with an increase in the likelihood of starting a career in STEM. Both coefficients are highly statistically significant. Interestingly, the magnitude of these coefficients is similar to the raw correlation between the variables and the outcome as provided by columns 1a and 1b.

In columns 1c and 2a, two further relationships emerge: First, we find no differences in choosing a STEM career among students of different migration backgrounds. Second, teenagers who have at least one parent who works in a STEM field are more likely to choose a STEM career themselves.

These findings remain consistent when using the WLE scores provided by the NEPS instead of our calculated mean score of ICT skills (see S4 Table). When using the WLE scores, the magnitude of the correlation can not be translated into the number of correctly answered questions. Instead, we can only state that above average ICT skills positively correlate with choosing a STEM career.

## ICT skills as a moderator

In columns 1d and 2b we add the interaction of a teenager's gender and ICT skills to the regression (as provided by Eq 3). The results are striking for teenagers in 9th grade: While females rather do not choose a career in STEM, those girls with higher digital competencies are much more likely to choose either a vocational training or an undergraduate college course in STEM. The coefficient of the interaction in 9th grade is 0.258. In combination with the coefficients for ICT skills (0.037 but lacking statistical significance) and female (-0.519, highly statistically significant), the interpretation is that an increase in girls' ICT skills by 10 percentage points is associated with an increase in the probability to choose a STEM career by 24.99%. The direction of the relationship of interest holds even if we control for mathematical skills in column 1e and when interacting mathematical skills with the variable for gender in column 1f. Correctly solving an additional three to four questions is still associated with an increase in the likelihood of choosing a STEM career by 21.71%. Again, the results are consistent when utilizing the WLE-scores though the variable ICT skills remains statistically significant throughout all specifications (columns 1d to 1f in S4 Table) while the interaction of ICT skills and gender is still positive but lacks statistical significance in column 1f of S4 Table. Overall, using the WLE score also points into the direction that ICT skills moderate the gender-specific sorting into STEM on top of mathematical skills.

For teenagers in grade 12 the relationship is different: While the direct relationship of ICT skills and gender remains high in magnitude and strong in statistical significance, the interaction of the two variables lacks economic and statistical significance. That means that the positive link between ICT skills and the choice of a career in STEM holds for female and male teenagers and no longer for female teenagers only. We explore this finding further in the next section.

When accounting for mathematical skills, the isolated effect of the digital skills slightly decreases (column 2c) and lacks statistical significance (column 2d), while at the same time the mathematical skills seem to be more relevant for subsequent educational choices than in ninth grade. Selection into more complex courses in high school as determined by [1] may be a driver of the link between skills and the choice of a STEM career. Using the WLE scores instead of our calculated mean scores provides similar results with improved statistical significance.

Overall, the results in Table 2 show that ICT skills rather act as a moderator in the career choice of girls in ninth grade while ICT skills and a student's gender rather independently affects career paths after 12th grade.

## Type of STEM training

Given these results, we dig deeper into the types of training in a second set of regressions on the subsample of students in grade 9. After having completed secondary schooling, students most commonly either start with an apprenticeship (vocational training) or go to college. We

**Table 3. Heterogeneity across type of STEM training.**

| | Dependent variable: | | | | | | | | |
|---|---|---|---|---|---|---|---|---|---|
| | Respondent chooses STEM occupation after high school | | | | | | | | |
| | Pooled | | | Vocational training | | | College | | |
| | Pooled | Female | Male | Pooled | Female | Male | Pooled | Female | Male |
| | (1a) | (1b) | (1c) | (2a) | (2b) | (2c) | (3a) | (3b) | (3c) |
| ICT skills in 9th grade | −0.016 | 0.187*** | −0.020 | −0.015 | 0.180*** | −0.020 | 0.055 | 0.002 | 0.056 |
| | (0.041) | (0.034) | (0.048) | (0.048) | (0.037) | (0.059) | (0.084) | (0.075) | (0.093) |
| ICT skills * Female | 0.194*** | - | - | 0.181** | - | - | −0.050 | - | - |
| | (0.059) | - | - | (0.071) | - | - | (0.117) | - | - |
| Female | −0.496*** | - | - | −0.512*** | - | - | −0.247*** | - | - |
| | (0.031) | - | - | (0.036) | - | - | (0.075) | - | - |
| Migration background | −0.008 | 0.004 | −0.022 | −0.024* | −0.009 | −0.039* | 0.019 | 0.014 | 0.025 |
| | (0.011) | (0.012) | (0.018) | (0.013) | (0.013) | (0.022) | (0.020) | (0.025) | (0.033) |
| Parent(s) in STEM occupation | 0.074*** | 0.051*** | 0.096*** | 0.071*** | 0.053*** | 0.086*** | 0.070*** | 0.051** | 0.093*** |
| | (0.009) | (0.011) | (0.015) | (0.011) | (0.011) | (0.019) | (0.017) | (0.021) | (0.028) |
| Mathematical skills | 0.038*** | 0.083*** | 0.036** | 0.047*** | 0.052*** | 0.045** | 0.054 | 0.057** | 0.052 |
| | (0.015) | (0.012) | (0.017) | (0.017) | (0.013) | (0.020) | (0.036) | (0.024) | (0.040) |
| Mathematical skills * Female | 0.043** | - | - | 0.004 | - | - | 0.002 | - | - |
| | (0.021) | - | - | (0.026) | - | - | (0.044) | - | - |
| Constant | 0.476*** | −0.021 | 0.475*** | 0.486*** | −0.031 | 0.488*** | 0.400*** | 0.161*** | 0.391*** |
| | (0.022) | (0.020) | (0.026) | (0.025) | (0.020) | (0.031) | (0.056) | (0.050) | (0.064) |
| Observations | 9,315 | 4,668 | 4,647 | 5,895 | 2,802 | 3,093 | 3,197 | 1,759 | 1,438 |
| R² | 0.174 | 0.039 | 0.010 | 0.225 | 0.033 | 0.011 | 0.097 | 0.007 | 0.011 |
| Adjusted R² | 0.173 | 0.038 | 0.010 | 0.224 | 0.032 | 0.009 | 0.095 | 0.005 | 0.008 |

The dependent variable is a binary variable that equals 1 if the respondents' longest training period within five years after completing secondary schooling was within a STEM field. Robust standard errors are clustered on the levels of schools. The data of 107 female and 216 male respondents has been dropped in columns (2) and (3) because we could not clearly define the training as vocational or college studies.
Significance:
*p<0.1;
**p<0.05;
***p<0.01.

run our regressions from Eqs 2 and 3 separately on different subsamples depending on gender and the training track. Table 3 presents the results.

Column 1a provides the baseline results (as in column 1f in Table 2). Columns 1b and 1c provide the results after running the pooled regressions separately for female (1b) and male students (1c). As expected, the association between female teenagers' ICT skills and choosing a STEM career is large in magnitude and highly statistically significant—while there is no clear pattern for male students.

Columns 1a, 2a, and 3a of Table 3 provide that female students are less likely to choose a STEM career. Furthermore, it appears that all students are affected by their parents' occupations. The relationship is stronger for male than for female students though: Male teenagers are more likely to choose any STEM career if at least one parent was working in STEM (columns 1c, 2c, and 3c).

Our main variable of interest is a respondent's ICT skills. Looking at the relationship between the ICT skills and the probability to choose a STEM career, Table 2 shows that the

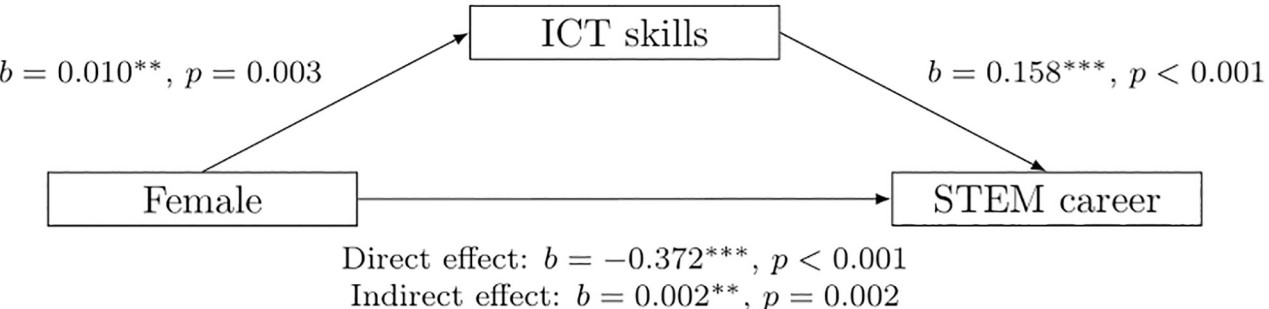

**Fig 3. Results from mediation analysis.** The figure illustrates the results of running the mediation analysis. Additional covariates are teenagers' migrant background and whether at least one of their parents works in a STEM occupation (as in column 1d of Table 2). Standard errors of the indirect effect are bootstrapped with 1,000 replications. Sample sizes are 9,315 for the pooled mediation (Panel A), 3,197 for the subsample of teenagers who start a vocational training (Panel B), and 5,895 for the subsample of college students (Panel C). Detailed regression tables are available on request.

strong association from our baseline regression disappears when conditioning on college education. All our results from column 1f in Table 2 seem to be driven by teenagers who start vocational training after completing high school. It is worth noting that the coefficient is large in magnitude and strong in statistical significance despite the small subset of only 2,802 female students who decide for a vocational training.

Summing this up, the preceding analysis suggests that teenagers' ICT skills correlate with their subsequent career choices (RQ 1): teenagers with higher ICT skills during secondary education are more likely to start a career in STEM. We also show that female teenagers select out of STEM careers. Regarding the interrelationship between a teenager's gender, ICT skills, and choice of STEM education, we can conclude that ICT skills act as a moderator for those who start a vocational training: Female teenagers are more likely to start an apprenticeship in a STEM field if they already have high ICT skills in 9th grade. Such a relationship neither exists for studying a STEM field at college nor for teenagers' ICT skills in 12th grade (RQ2).

**Mediation analysis.** In the preceding results, we have already identified a joint action of gender and ICT skills. In this section, we now turn to the results of a mediation analysis (RQ 3). We used the *mediation* package provided by [49] for R to conduct the mediation analysis. The results summarised in Fig 3 reveal that the gender-specific selection into STEM fields is not mediated via the teenagers' ICT skills.

The results in Fig 3 imply that the gender-specific sorting into STEM fields is not caused by gender-specific development of ICT skills. We find a strong negative direct effect of being female on the choice of a STEM field ($b = -0.286$ for pooled analysis in panel A, $b = -0.286$ for vocational training in panel B, $b = -0.426$ for college in panel C) and a positive though not always statistically significant effect of having larger ICT skills on choosing a STEM field ($b = 0.158$ for pooled, $b = 0.078$ for vocational training, $b = 0.119$ for college). The indirect effect is negligibly small and even lacks statistical significance for the subsamples that include only those teenagers who start vocational training (Panel B) or who go to college (Panel C). Overall, the mediation analysis shows that there is no causal pathway along which a teenager's gender causes their ICT skills (the mediator) and the ICT skills cause the choice of a career in STEM.

## Discussion and robustness

The results provide insights into the relationship between teenagers' gender, ICT skills, and career choices. First, higher ICT skills do not per se translate into a career in STEM. We find,

however, very strong positive associations between girls' ICT skills in ninth grade and their decisions to start a vocational training in a STEM field. Interestingly, ICT skills act as a moderator and not as a mediator in the triangle of interest: Male teenagers choose a STEM career upon graduation from secondary school independent of their ICT skills in ninth grade. In turn, female teenagers choose a STEM career if they exert above average ICT skills in grade 9. The different empirical approaches revealed these relationships and provide fertile ground for future research and policymaking.

One explanation for the moderating effect of ICT skills on the gender-specific sorting into STEM comes from research in the field of Psychology: [5] have shown that young people are interested in occupations that imply tasks they believe to be good at while [6–8] show that female teenagers tend to underestimate their true potential. Combining these two strands may imply that female teenagers' potential lack of confidence in ICT ability may hinder them from starting a career in STEM. In that respect, girls' self-perception regarding their ICT skills may be similar to that of their mathematical skills: [13, 14] show that self-perceived mathematical abilities predict choosing maths at high school and college even after controlling for actual mathematical abilities.

Our findings add to existing evidence by [50, 51]: Based on a sample of 1,386 children from lower secondary school, [50] find that internet self-efficacy affects STEM performance more than actual ICT readiness. They suggest that teachers should aim for improving students' self-efficacy. Similarly, [51] evaluate a local ICT training workshop for 411 female students from lower secondary school and conclude that gender stereotyping in ICT can be reduced by granting more learning opportunities to female teenagers.

Another relevant finding of the present study is that higher ICT skills close to receiving a university entrance qualification come with a stronger selection into studying STEM. This relationship as presented in Table 2 holds for female and male teenagers. One of the underlying reasons may be that students in high schools have to specialize in some fields and the overall selection into or out of STEM at least partly already takes place in upper secondary schooling as shown by [1] and also by [9]. When students choose more STEM subjects they are likely to increase their ICT skills at school by attending more ICT-related courses (e.g., in Physics) and also outside school because they are more interested in improving their skills. The finding shows the importance to reduce gender-stereotyping in ICT as early as lower secondary school onwards, i.e., before actual gender gaps in ICT skills emerge.

## Robustness I: Changes in the definition of the training period

So far, we have concentrated on the longest training period within five years after completing secondary schooling. However, students may change their desired career within the first years after graduating from high school.

The aforementioned results as presented in Table 2 use the *longest* training period as the outcome variable. We believe that the longest training period best captures the long-term career choice. To understand whether different training periods affect the results, we run our baseline regression from Eq 2 but use the *first* training period after completing high school as well as the *last* training period within the five years after high school instead of the longest training period.

S3 Table provides the results of the alternative specification of the outcome variable. Column (1) provides the baseline specification (as in column 1f in Table 2). Overall, the results are very similar, indicating—as already suggested by Table 1—that students are very consistent in choosing either STEM or non-STEM even if they change the exact subfield or switch their training provider.

An interesting difference between the three specifications in S3 Table is, however, the decrease in importance of female teenagers' ICT skills when considering the last training period (column 3) while at the same time, the mathematical skills increase in their importance. The ad-hoc selection into STEM appears to be (partly) driven by teenagers' ICT skills.

Overall, it appears that the results are very robust to considering the different timing of the training. The ICT skills remain highly correlated with the decision towards a career in STEM —but for female students only.

### Robustness II: Selectivity of sample

NEPS data is representative of German high school students. However, not all students participated in the ICT skills test. S1 Table compares our sample to the full data and shows that our sample is slightly positive selected in terms of ICT skills, overall mathematical abilities, and parents working in STEM fields. In turn, the sample comprise fewer students from migrant families than the full data.

Overall, the sample is slightly positively selected and is thereby no longer fully representative of German teenagers: teenagers in the sample live in households with above-average parental job engagement in STEM and below-average migrant backgrounds. We believe that the students in the sample have a higher than population-average probability to choose a career in STEM because their parents can serve as role models. Thereby, we may underestimate the true (population-wide) relationship between teenagers' digital skills and their subsequent choice of a STEM occupation.

## Conclusion and policy recommendations

The present study aims at understanding whether the level of teenagers' ICT skills affects their career choices and whether the gender-specific sorting into STEM fields can be explained by differences in ICT skills. The study is based on data from the National Educational Panel Study (NEPS) that comprises detailed data about individual educational trajectories and also provides test-based results on teenagers' ICT skills. We use a sample of 9,315 teenagers who were in ninth grade in the school year 2010/11 and were repeatedly asked about their educational choices and pathways afterwards. Tests on students' ICT skills were conducted in 2010/2011 and again in 2013/14. We look at the transition into vocational training or higher education after high school.

While we find—in line with recent evidence by [25, 52, 53]—no significant differences between girls' and boys' ICT skills in ninth grade, the subsequent trajectories of female and male teenagers are different: Only those girls with strong ICT skills select into an apprenticeship in STEM. A student's ICT skills act as a moderator in the the gender-specific selection into STEM. The relationship between an indvidual's ICT skills and subsequent career choices is, however, different among students in their final years of high school: For students in 12th grade, higher ICT skills (and higher mathematical skills) directly affect the choice of a STEM field while we could neither find a moderating nor a mediating effect of ICT skills on the choice of a STEM field.

Our results complement recent evidence by [1, 9] who show that the persistent gender gap in STEM is partly caused by some students' lack of engagement in complex STEM fields in high school. The present study extends on this line of research by indicating that the behaviour is at least partly driven by ICT skills and that girls need above average ICT skills to sort into STEM fields upon graduating from high school. We explain our findings with existing evidence that shows that teenagers sort into occupations they believe to be good at [5] and that female teenagers rather underestimate their true potential [6–8]. Moreover, research on

mathematical competencies shows that even female college students lack confidence in their mathematical abilities even if they exert the same competence level as their male fellow students [54].

The shortage in the supply of STEM workers is a major concern in many countries and industries [2]. Given the persistent gender-specific selection into STEM fields, our study contributes to a set of studies that aim at understanding the lack of female participation in STEM occupations. While research has shown that, for instance, teachers and role models can reduce stereotypical sorting into jobs [55], the present study shows that a (perceived) lack of ICT skills may hinder girls from choosing a STEM field after high school.

Thereby, the findings of the present study have strong policy implications: Promoting ICT skills and strengthening confidence in ICT, especially among female teenagers during their secondary schooling, can increase the number of teenagers starting a career in STEM and thereby combat the persistent gender gap in STEM. [56] show that self-efficacy in computing among underrepresented groups at college can be increased in special courses and mentoring programs. [51] evaluate an ICT training course for high school girls and also show that self-efficacy can be increased and gender-stereotypes reduced in specific trainings. Thereby, the findings of this study add to the ongoing policy debate on increasing the intensity of ICT training in secondary school [27, 28].

Finally, building up ICT skills is not only relevant for reducing skill shortage in STEM or closing the gender gap in STEM. [57] show that ICT skills can help to protect workers against displacement risk stemming from skill-biased technological change and the use of robots and artificial intelligence. In the ever-evolving digital world, low ICT skills can lead to limited career options and lower incomes in the long run [58].

## Supporting information

**S1 Table. Comparison of sample with full data.**
(ZIP)

**S2 Table. Descriptive statistics of students tested in grade 12.**
(ZIP)

**S3 Table. Alternative definitions of outcome in main regression.**
(ZIP)

**S4 Table. ICT skills in 9th and 12th grade and career paths: Direct relationship and moderating effect (using WLE scores).**
(ZIP)

**S1 Fig. Illustration of all variables of regressions.** S1 Fig. illustrates the empirical analysis including all variables. In addition to ICT skills and gender, we also include a teenager's migration background as well as their mathematical skills. As we do not know a priori whether ICT (or mathematical) skills moderate the gender-specific selection into STEM, we also include interaction terms in some specifications as indicated by the dashed arrow. Please note that Fig 2 in the main text has concentrated on the main variables of interest—gender, ICT skills, and choice of a STEM career.
(PNG)

## Acknowledgments

The authors thank Ronald Bachmann for fruitful discussions and valuable feedback. They also thank Qaisar Abbas, Endale Tadesse, Laurice Tolentino, and an anonymous referee for their

comments. Heike Ermert, Sarah Hoene and Rachel Kühn provided excellent research assistance.

## Author Contributions

**Conceptualization:** Judith Lehner.

**Data curation:** Friederike Hertweck, Judith Lehner.

**Formal analysis:** Friederike Hertweck, Judith Lehner.

**Methodology:** Friederike Hertweck, Judith Lehner.

**Project administration:** Friederike Hertweck.

**Writing – original draft:** Friederike Hertweck, Judith Lehner.

**Writing – review & editing:** Friederike Hertweck.

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
