## [Decision Letter · Decision Letter 0]

26 Dec 2023

PONE-D-23-30220The Gender Gap in STEM: (Female) Teenagers' ICT skills and subsequent career pathsPLOS ONE

Dear Dr. Hertweck,

Thank you for submitting your manuscript to PLOS ONE. After careful consideration, we feel that it has merit but does not fully meet PLOS ONE’s publication criteria as it currently stands. Therefore, we invite you to submit a revised version of the manuscript that addresses the points raised during the review process.

**Please revise article according to **Reviewers' comments mentioned below:

We look forward to receiving your revised manuscript.

Kind regards,

Qaisar Abbas, Ph.D

Academic Editor

PLOS ONE

Reviewers' comments:

Reviewer's Responses to Questions

**Comments to the Author**

1. Is the manuscript technically sound, and do the data support the conclusions?

Reviewer #1: Yes

Reviewer #2: Yes

Reviewer #3: Yes

2. Has the statistical analysis been performed appropriately and rigorously? 

Reviewer #1: Yes

Reviewer #2: Yes

Reviewer #3: Yes

3. Have the authors made all data underlying the findings in their manuscript fully available?

Reviewer #1: Yes

Reviewer #2: Yes

Reviewer #3: Yes

4. Is the manuscript presented in an intelligible fashion and written in standard English?

Reviewer #1: Yes

Reviewer #2: Yes

Reviewer #3: Yes

5. Review Comments to the Author

Reviewer #1: Dear authors,

I have read your paper, titled " The Gender Gap in STEM: (Female) Teenagers' ICT skills and subsequent career paths ", carefully.

I am of the convection that the paper needs some serious revisions before I recommend it for publication in this journal. Please necessarily apply the comments and highlight them in the paper.

The following presents my comments.

1.The Abstract section of the paper has been written very poorly. You are expected to state the necessity of your research and its novelty correctly. Please describe the software implementation and algorithm results briefly. I think the Abstract section needs to be 200-300 words and formatted in a standard way. Please revise the Abstract.

2.Provide a new headline at the end of the paper. In this section, the output results of the paper should be compared with related references numerically or graphically. In this section, the superiority of the algorithm presented in this paper should be clarified compared to other literature. It is better to define some indices for this comparison.

3.Describe your algorithm steps as a flowchart or a flow graph. The input and output of the algorithm must be specified. The absence of this flowchart confuses readers.

4.Please use more papers similar to your work that have been published in this specific journal.

Reviewer #2: It is a great pleasure to read such a great work that the authors put a lots of effort on it. In the meantime, I have stated a couple of points need to be addressed before the manuscript is ready for publication.

1. The authors need to choose either female or teenagers/ adolescents

2. The discussion part lacks knitting existing literature and findings in different context to comparing it with the present study finding.

Reviewer #3: After the Conclusions, write a summary of recommendations about the understanding of whether teenage ICT proficiency influences career choices and whether gender inequalities in ICT proficiency can account for the gendered sorting into STEM abilities.

6. PLOS authors have the option to publish the peer review history of their article (what does this mean?). If published, this will include your full peer review and any attached files.

Reviewer #1: No

Reviewer #2: **Yes: **Endale Tadesse

Reviewer #3: **Yes: **Ms. Laurice Tolentino

---

## [Author Response · Author response to Decision Letter 0]

11 Jan 2024

### Reply to Referee 1

Authors: We thank the referee for his or her helpful comments. These are particularly helpful given the interdisciplinary nature of the journal. In the following, we explain how we have dealt with the comments.

Comment 1: 

The Abstract section of the paper has been written very poorly. You are expected to state the necessity of your research and its novelty correctly. Please describe the software implementation and algorithm results briefly. I think the Abstract section needs to be 200-300 words and formatted in a standard way. Please revise the Abstract.

Reply:

We have revised the abstract. We have now included additional sentences explaining the necessity and novelty of our research. Moreover, we describe the results in more detail.

Additional sentences regarding the necessity: 

• “To overcome shortage of STEM talent, the selection into STEM fields must be fully understood.”

• “By addressing the interplay between gender, ICT skills, and educational choices, the present study uncovers an additional lever of how to mitigate skills shortage in STEM.”

Additional sentence regarding the novelty: 

• “We contribute to existing research on the selection of STEM careers by analysing the interplay between teenagers' proficiency in Information and Communication Technology (ICT) and their career preferences in the STEM domain.”

Additional description of results: 

• “An increase in girls' ICT skills by 10 percentage points in ninth grade is associated with an increase in the probability to choose a STEM career by 2.95 percentage points.”

New word count of the abstract: 220 words.

Comment 2: 

Provide a new headline at the end of the paper. In this section, the output results of the paper should be compared with related references numerically or graphically. In this section, the superiority of the algorithm presented in this paper should be clarified compared to other literature. It is better to define some indices for this comparison. 

Reply:

We have extended the “discussion” and “conclusion” sections to better compare our findings with existing literature and to present why our empirical strategy helped revealing the findings. In particular, we included the following paragraphs in the discussion section:

“(…) Interestingly, ICT skills act as a moderator and not as a mediator in the triangle of interest. This finding is of interest for policymakers and researchers alike: Policymakers need to understand the relevant age range when to improve teenagers' ICT skills. The present study points into the direction that male teenagers choose a STEM career upon graduation from secondary school independent of their ICT skills in ninth grade. In turn, female teenagers only choose a STEM career if they exert above average ICT skills in grade 9. The different empirical approaches revealed these relationships and provide fertile ground for future research.” 

In the conclusion, we added:

“Moreover, research on mathematical competencies shows that even female college students lack confidence in their mathematical abilities even if they exert the same competence level as

their male fellow students (Ellis, Fosdick, & Rasmussen, 2016).

(…)

Thereby, the findings of the present study have strong policy implications: Promoting ICT skills and strengthening confidence in ICT, especially among female teenagers during their secondary schooling, can increase the number of teenagers starting a career in STEM and thereby combat the persistent gender gap in STEM. Durham Brooks, Burks, Doyle, Meysenburg, and Frey (2021) show that self-efficacy in computing among underrepresented groups at college can be increased in special courses and mentoring programms. Tam et al. (2020) evaluate an ICT training course for high school girls and also shows that self-efficacy can be increased and gender-stereotypes reduced in specific trainings. Thereby, the findings of this study add to the ongoing policy debate on increasing the intensity of ICT training in secondary school (Vegas et al., 2021; KMK, 2016).

Finally, building up ICT skills is not only relevant for reducing skill shortage in STEM or closing the gender gap in STEM. Chen, Li and Tang (2022) show that ICT skills can help to protect workers against displacement risk stemming from skill-biased technological change and the use of robots and artificial intelligence. In the ever-evolving digital world, low ICT skills can lead to limited career options and lower incomes in the long run (Falck, Heimisch-Roecker, & Wiederhold, 2021).”

We decided not to include an additional heading but to integrate the additional paragraphs in the existing sections spanning the discussion and the conclusion.

Overall, we added the following literature to the discussion and the results section:

• Lu, C., Yang, W., Wu, L. et al. How Behavioral and Psychological Factors Influence STEM Performance in K-12 Schools: A Mediation Model. J Sci Educ Technol 32, 379–389 (2023). https://doi.org/10.1007/s10956-023-10034-3

• Tam, H.-L., Chan, A. Y.-F., & Lai, O. L.-H. (2020). Gender stereotyping and STEM education: Girls’ empowerment through effective ICT training in Hong Kong. Children and Youth Services Review, 119, 105624. https://doi.org/10.1016/j.childyouth.2020.105624

• Bachmann, R., & Hertweck, F. (2023). The gender gap in digital literacy: a cohort analysis for Germany. Applied Economics Letters, 0 (0), 1-6. https://doi.org/10.1080/13504851.2023.2277685

• Durham Brooks, T., Burks, R., Doyle, E., Meysenburg, M., & Frey, T. (2021). Digital imaging and vision analysis in science project improves the self-efficacy and skill of undergraduate students in computational work. PLOS ONE, 16 (5), e0241946. https://doi.org/10.1371/journal.pone.0241946.

• Ellis, J., Fosdick, B. K., & Rasmussen, C. (2016). Women 1.5 times more likely to leave STEM pipeline after calculus compared to men: Lack of mathematical confidence a potential culprit. PLOS ONE, 11 (7), e0157447. https://doi.org/10.1371/journal.pone.0157447

• Falck, O., Heimisch-Roecker, A., & Wiederhold, S. (2021). Returns to ICT skills. Research Policy, 50 (7), 104064. https://doi.org/10.1016/j.respol.2020.104064

Comment 3:

Describe your algorithm steps as a flowchart or a flow graph. The input and output of the algorithm must be specified. The absence of this flowchart confuses readers. 

Reply:

PLOS One is an interdisciplinary journal and we highly appreciate your comment. We believe our work is relevant for various disciplines (that’s why we chose the journal) and aimed for displaying the empirical strategy and the results in a way that can be understood by researchers from various disciplines. We understand that not including all variables in Figure 2 may have caused the confusion you are mentioning. So we are glad you pointed it out.

We added Figure 4 to the appendix that illustrates all variables that have been employed in the empirical analysis. We decided to not include the full chart in the main text as Figure 2 already illustrates the key elements of the algorithm and the empirical approaches we follow. In the main body of the text, we would like to emphasize the three ways on modelling the interplay between gender, ICT skills and STEM career – because this is exactly where we add to existing evidence. Hence, we would like to keep the figures in the main body of the text as parsimonious as possible but added Figure 4 to the appendix. 

Comment 4:

Please use more papers similar to your work that have been published in this specific journal.

Reply:

We have have added two relevant papers from PLOS ONE:

• Chen, N., Li, Z., & Tang, B. (2022). Can digital skill protect against job displacement risk caused by artificial intelligence? empirical evidence from 701 detailed occupations. PLOS ONE, 17 (11), e0277280.

• Ellis, J., Fosdick, B. K., & Rasmussen, C. (2016). Women 1.5 times more likely to leave stem pipeline after calculus compared to men: Lack of mathematical confidence a potential culprit. PLOS ONE, 11 (7), e0157447.

#### Referee 2:

Authors: We thank the referee for his helpful comments. In the following, we explain how we have dealt with the comments.

Comment 1:

The authors need to choose either female or teenagers/ adolescents.

Reply:

We now stick to “teenagers” throughout the article. 

Comment 2: 

The discussion part lacks knitting existing literature and findings in different context to comparing it with the present study finding.

Reply:

We added two paragraphs to the discussion section. Regarding the literature, we added to the discussion section:

“Our findings add to existing evidence by Lu, Yang, Wu, and Yang (2023) and Tam, Chan, and Lai (2020): Based on a sample of 1,386 children from lower secondary school, Lu et al. (2023) find that internet self-efficacy affects STEM performance more than actual ICT readiness. They suggest that teachers should aim for improving students’ self-efficacy. Similarly, Tam et al. (2020) evaluate a local ICT training workshop for 411 female students from lower secondary school and conclude that gender stereotyping in ICT can be reduced by granting more learning opportunities to female teenagers.”

We also added the following literature to the results section. 

• Bachmann, R., & Hertweck, F. (2023). The gender gap in digital literacy: a cohort analysis for Germany. Applied Economics Letters, 0 (0), 1-6. doi: https://doi.org/10.1080/13504851.2023.2277685.

• Chen, N., Li, Z., & Tang, B. (2022). Can digital skill protect against job displacement risk caused by artificial intelligence? empirical evidence from 701 detailed occupations. PLOS ONE, 17 (11), e0277280.

• Ellis, J., Fosdick, B. K., & Rasmussen, C. (2016). Women 1.5 times more likely to leave stem pipeline after calculus compared to men: Lack of mathematical confidence a potential culprit. PLOS ONE, 11 (7), e0157447.

• Falck, O., Heimisch-Roecker, A., & Wiederhold, S. (2021). Returns to ICT skills. Research Policy, 50 (7), 104064. doi: https://doi.org/10.1016/j.respol.2020.104064.

### Referee 3:

Authors: We thank the referee for her helpful comments. In the following, we explain how we have dealt with the comments.

Comment 1:

After the Conclusions, write a summary of recommendations about the understanding of whether teenage ICT proficiency influences career choices and whether gender inequalities in ICT proficiency can account for the gendered sorting into STEM abilities.

Reply:

We have extended the conclusion part and added a paragraph with policy recommendations. Specifically, we added:

Thereby, the findings of the present study have strong policy implications: Promoting ICT skills and strengthening confidence in ICT, especially among female teenagers during their secondary schooling, can increase the number of teenagers starting a career in STEM and thereby combat the persistent gender gap in STEM. Durham Brooks, Burks, Doyle, Meysenburg, and Frey (2021) show that self-efficacy in computing among underrepresented groups at college can be increased in special courses and mentoring programs. Tam et al. (2020) evaluate an ICT training course for high school girls and also shows that self-efficacy can be increased and gender-stereotypes reduced in specific trainings. Thereby, the findings of this study add to the ongoing policy debate on increasing the intensity of ICT training in secondary school (Vegas et al., 2021; KMK, 2016).

Finally, building up ICT skills is not only relevant for reducing skill shortage in STEM or closing the gender gap in STEM. Chen, Li and Tang (2022) show that ICT skills can help to protect workers against displacement risk stemming from skill-biased technological change and the use of robots and artificial intelligence. In the ever-evolving digital world, low ICT skills can lead to limited career options and lower incomes in the long run (Falck, Heimisch-Roecker, & Wiederhold, 2021).”

---

## [Decision Letter · Decision Letter 1]

17 Jul 2024

The Gender Gap in STEM: (Female) Teenagers' ICT skills and subsequent career paths

PONE-D-23-30220R1

Dear Dr. Hertweck,

We’re pleased to inform you that your manuscript has been judged scientifically suitable for publication and will be formally accepted for publication once it meets all outstanding technical requirements.

Kind regards,

Najmul Hasan, PhD

Academic Editor

PLOS ONE

Additional Editor Comments (optional):

Reviewers' comments:

Reviewer's Responses to Questions

**Comments to the Author**

1. If the authors have adequately addressed your comments raised in a previous round of review and you feel that this manuscript is now acceptable for publication, you may indicate that here to bypass the “Comments to the Author” section, enter your conflict of interest statement in the “Confidential to Editor” section, and submit your "Accept" recommendation.

Reviewer #2: All comments have been addressed

2. Is the manuscript technically sound, and do the data support the conclusions?

Reviewer #2: Yes

3. Has the statistical analysis been performed appropriately and rigorously? 

Reviewer #2: Yes

4. Have the authors made all data underlying the findings in their manuscript fully available?

Reviewer #2: Yes

5. Is the manuscript presented in an intelligible fashion and written in standard English?

Reviewer #2: Yes

6. Review Comments to the Author

Reviewer #2: Thank you for addressing every inquiry mentioed in the earlier comment. Good luck with the publication.

7. PLOS authors have the option to publish the peer review history of their article (what does this mean?). If published, this will include your full peer review and any attached files.

Reviewer #2: **Yes: **Endale Tadesse

---

## [Editor Report · Acceptance letter]

23 Sep 2024

PONE-D-23-30220R1 

PLOS ONE

Dear Dr. Hertweck, 

I'm pleased to inform you that your manuscript has been deemed suitable for publication in PLOS ONE. Congratulations! Your manuscript is now being handed over to our production team.

Kind regards, 

on behalf of

Dr. Najmul Hasan 

Academic Editor

PLOS ONE